# MODEL DISTILLATION WITH KNOWLEDGE TRANSFER FROM FACE CLASSIFICATION TO ALIGNMENT AND VERIFICATION

## ABSTRACT

Knowledge distillation is a potential solution for model compression. The idea is to make a small student network imitate the target of a large teacher network, then the student network can be competitive to the teacher one. Most previous studies focus on model distillation in the classification task, where they propose different architectures and initializations for the student network. However, only the classification task is not enough, and other related tasks such as regression and retrieval are barely considered. To solve the problem, in this paper, we take face recognition as a breaking point and propose model distillation with knowledge transfer from face classification to alignment and verification. By selecting appropriate initializations and targets in the knowledge transfer, the distillation can be easier in non-classification tasks. Experiments on the CelebA and CASIA-WebFace datasets demonstrate that the student network can be competitive to the teacher one in alignment and verification, and even surpasses the teacher network under specific compression rates. In addition, to achieve stronger knowledge transfer, we also use a common initialization trick to improve the distillation performance of classification. Evaluations on the CASIA-Webface and large-scale MS-Celeb-1M datasets show the effectiveness of this simple trick.

## 1 INTRODUCTION

Since the emergence of Alexnet(Krizhevsky et al., 2012), larger and deeper networks have shown to be more powerful(Simonyan & Zisserman, 2015; Szegedy et al., 2015; He et al., 2016). However, as the network going larger and deeper, it becomes difficult to use it in mobile devices. Therefore, model compression has become necessary in compressing the large network into a small one. In recent years, many compression methods have been proposed, including knowledge distillation(Ba & Caruana, 2014; Hinton et al., 2014; Romero et al., 2015), weight quantization(Gong et al., 2015; Rastegari et al., 2016), weight pruning(Han et al., 2016; Szegedy et al., 2016) and weight decomposition(Canziani et al., 2017; Novikov et al., 2015). In this paper, we focus on the knowledge distillation, which is a potential approach for model compression.

In knowledge distillation, there is usually a large teacher network and a small student one, and the objective is to make the student network competitive to the teacher one by learning specific targets of the teacher network. Previous studies mainly consider the selection of targets in the classification task, *e.g.,* hidden layers(Luo et al., 2016), logits(Ba & Caruana, 2014; Urban et al., 2017; Sau & Balasubramanian, 2017) or soft predictions(Hinton et al., 2014; Romero et al., 2015). However, only the distillation of the classification task is not enough, and some common tasks such as regression and retrieval should also be considered. In this paper, we take face recognition as a breaking point that we start with the knowledge distillation in face classification, and consider the distillation on two domain-similar tasks, including face alignment and verification. The objective of face alignment is to locate the key-point locations in each image; while in face verification, we have to determine if two images belong to the same identity.

For distillation on non-classification tasks, one intuitive idea is to adopt a similar method as in face classification that trains teacher and student networks from scratch. In this way, the distillation on all tasks will be independent, and this is a possible solution. However, this independence cannot

give the best distillation performance. There has been strong evidence that in object detection(Ren et al., 2015), object segmentation(Chen et al., 2015) and image retrieval(Zhao et al., 2015), they all used the pretrained classification model(on ImageNet) as initialization to boost performance. This success comes from the fact that their domains are similar, which makes them transfer a lot from low-level to high-level representation(Yosinski et al., 2014). Similarly, face classification, alignment and verification also share the similar domain, thus we propose to transfer the distilled knowledge of classification by taking its teacher and student networks to initialize corresponding networks in alignment and verification.

Another problem in knowledge transfer is what targets should be used for distillation? In face classification, the knowledge is distilled from the teacher network by learning its soft-prediction, which has been proved to work well(Hinton et al., 2014; Romero et al., 2015). However, in face alignment(Wu et al., 2015) and verification(Wu et al., 2015), they have additional task-specific targets. As a result, selecting the classification or task-specific target for distillation remains a problem. One intuitive idea is to measure the relevance of objectives between non-classification and classification tasks. For example, it is not obvious to see the relation between face classification and alignment, but the classification can help a lot in verification. Therefore, it seems reasonable that if the tasks are highly related, the classification target is preferred, or the task-specific target is better.

Inspired by the above thoughts, in this paper, we propose the model distillation in face alignment and verification by transferring the distilled knowledge from face classification. With appropriate selection of initializations and targets, we show that the distillation performance of alignment and verification on the CelebA(Liu et al., 2015) and CASIA-WebFace(Yi et al., 2016) datasets can be largely improved, and the student network can even exceed the teacher network under specific compression rates.

This knowledge transfer is our main contribution. In addition, we realize that in the proposed method, the knowledge transfer depends on the distillation of classification, thus we use a common initialization trick to further boost the distillation performance of classification. Evaluations on the CASIA-WebFace and large-scale MS-Celeb-1M(Guo et al., 2016) datasets show that this simple trick can give the best distillation results in the classification task.

## 2 RELATED WORK

In this part, we introduce some previous studies on knowledge distillation. Particularly, all the following studies focus on the classification task. Buciluă et al. (2006) propose to generate synthetic data by a teacher network, then a student network is trained with the data to mimic the identity labels. However, Ba & Caruana (2014) observe that these labels have lost the uncertainties of the teacher network, thus they propose to regress the logits (pre-softmax activations)(Hinton et al., 2014). Besides, they prefer the student network to be deep, which is good to mimic complex functions. To better learn the function, Urban et al. (2017) observe the student network should not only be deep, but also be convolutional, and they get competitive performance to the teacher network in CIFAR(Krizhevsky & Hinton, 2009). Most methods need multiple teacher networks for better distillation, but this will take a long training and inference time(Sau & Balasubramanian, 2017). To address the issue, Sau & Balasubramanian (2017) propose noise-based regularization that can simulate the logits of multiple teacher networks. However, Luo et al. (2016) observe the values of these logits are unconstrained, and the high dimensionality will cause fitting problem. As a result, they use hidden layers as they capture as much information as the logits but are more compact.

All these methods only use the targets of the teacher network in distillation, while if the target is not confident, the training will be difficult. To solve the problem, Hinton et al. (2014) propose a multi-task approach which uses identity labels and the target of the teacher network jointly. Particularly, they use the post-softmax activation with temperature smoothing as the target, which can better represent the label distribution. One problem is that student networks are mostly trained from scratch. Given the fact that initialization is important, Romero et al. (2015) propose to initialize the shallow layers of the student network by regressing the mid-level target of the teacher network. However, these studies only consider knowledge distillation in classification, which largely limits its application in model compression. In this paper, we consider face recognition as a breaking point and extend knowledge distillation to non-classification tasks.

## 3 DISTILLATION OF CLASSIFICATION

Due to the proposed knowledge transfer depends on the distillation of classification, improving the classification itself is necessary. In this part, we first review the idea of distillation for classification, then introduce how to boost it by a simple initialization trick.

### 3.1 REVIEW OF KNOWLEDGE DISTILLATION

We adopt the distillation framework in Hinton et al. (2014), which is summarized as follows. Let $T$ and $S$ be the teacher and student network, and their post-softmax predictions to be $\mathbf{P_T} = \mathrm{softmax}(\mathbf{a_T})$ and $\mathbf{P_S} = \mathrm{softmax}(\mathbf{a_S})$, where $\mathbf{a_T}$ and $\mathbf{a_S}$ are the pre-softmax predictions, also called the logits(Ba & Caruana, 2014). However, the post-softmax predictions have lost some relative uncertainties that are more informative, thus a temperature parameter $\tau$ is used to smooth predictions $\mathbf{P_T}$ and $\mathbf{P_S}$ to be $\mathbf{P_T^\tau}$ and $\mathbf{P_S^\tau}$, which are denoted as **_soft predictions_**:

$$\mathbf{P_T^\tau} = \mathrm{softmax}(\mathbf{a_T}/\tau), \quad \mathbf{P_S^\tau} = \mathrm{softmax}(\mathbf{a_S}/\tau). \tag{1}$$

Then, consider $\mathbf{P_T^\tau}$ as the target, knowledge distillation optimizes the following loss function

$$\mathrm{L}(\mathbf{W_S^{cls}}) = \mathrm{H}(\mathbf{P_S}, \mathbf{y^{cls}}) + \alpha\mathrm{H}(\mathbf{P_S^\tau}, \mathbf{P_T^\tau}), \tag{2}$$

wherein $\mathbf{W_S^{cls}}$ is the parameter of the student network, and $\mathbf{y^{cls}}$ is the identity label. For simplicity, we omit $min$ and the number of samples $N$, and denote the upper right symbol $cls$ as the classification task. In addition, $\mathrm{H}(,)$ is the cross-entropy, thus the first term is the softmax loss, while the second one is the cross-entropy between the soft predictions of the teacher and student network, with $\alpha$ balancing between the two terms. This multi-task training is advantageous because the target $\mathbf{P_T^\tau}$ cannot be guaranteed to be always correct, and if the target is not confident, the identity label $\mathbf{y^{cls}}$ will take over the training of the student work.

### 3.2 INITIALIZATION TRICK

It is noticed that in Eqn.(2), the student network is trained from scratch. As demonstrated in Ba & Caruana (2014) that deeper student networks are better for distillation, initialization thus has become very important(Hinton et al., 2006; Ioffe & Szegedy, 2015). Based on the evidence, Fitnet(Romero et al., 2015) first initializes the shallow layers of the student network by regressing the mid-level target of the teacher network, then it follows Eqn.(2) for distillation. However, only initializing the shallow layers is still difficult to learn high-level representation, which is generated by deep layers. Furthermore, Yosinski et al. (2014) shows that the network transferability increases as tasks become more similar. In our case, the initialization and distillation are both classification tasks with exactly the same data and identity labels, thus more deep layers should be initialized for higher transferability, and we use a simple trick to achieve this.

To obtain an initial student network, we train it with softmax loss:

$$\mathrm{L}(\mathbf{W_{S_0}^{cls}}) = \mathrm{H}(\mathbf{P_S}, \mathbf{y^{cls}}), \tag{3}$$

wherein the lower right symbol $S_0$ denotes the initialization for student network $S$. In this way, the student network is fully initialized. Then, we modify Eqn.(2) as

$$\mathrm{L}(\mathbf{W_S^{cls}}|\mathbf{W_{S_0}^{cls}}) = \mathrm{H}(\mathbf{P_S}, \mathbf{y^{cls}}) + \alpha\mathrm{H}(\mathbf{P_S^\tau}, \mathbf{P_T^\tau}), \tag{4}$$

wherein $\mathbf{W_S^{cls}}|\mathbf{W_{S_0}^{cls}}$ indicates that $\mathbf{W_S^{cls}}$ is trained with the initialization of $\mathbf{W_{S_0}^{cls}}$, and the two entropy terms remain the same. This process is shown in Fig.1(a). It can be seen that the only difference with Eqn.(2) is that the student network is trained with the full initialization, and this simple trick has been commonly used, *e.g.,* initializing the VGG-16 model based on a fully pretrained model(Simonyan & Zisserman, 2015). We later show that this trick can get promising improvements over Eqn.(2) and Fitnet(Romero et al., 2015).

## 4 DISTILLATION TRANSFER

In this part, we show how to transfer the distilled knowledge from face classification to face alignment and verification. The knowledge transfer consists of two steps: transfer initialization and target selection, which are elaborated as follows.

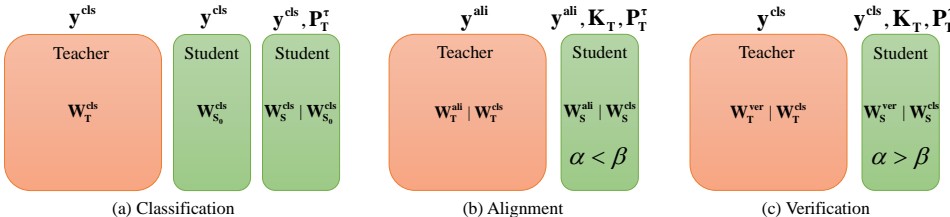

Figure 1: The pipeline of knowledge distillation in face classification, alignment and verification. $\mathbf{y}^{\mathbf{cls}}$ and $\mathbf{y}^{\mathbf{ali}}$ are the ground truth labels for classification and alignment respectively.

## 4.1 TRANSFER INITIALIZATION

The first step of the transfer is transfer initialization. The motivation is based on the evidence that in detection, segmentation and retrieval, they have used the pretrained classification model (on ImageNet) as initialization to boost performance(Ren et al., 2015; Chen et al., 2015; Zhao et al., 2015). The availability of this idea comes from the fact that they share the similar domain, which makes them transfer easily from low-level to high-level representation(Yosinski et al., 2014). Similarly, the domains of face classification, alignment and verification are also similar, thus we can transfer the distilled knowledge in the same way.

For simplicity, we denote the parameters of teacher and student networks in face classification as $\mathbf{W}_{\mathbf{T}}^{\mathbf{cls}}$ and $\mathbf{W}_{\mathbf{S}}^{\mathbf{cls}}$. Analogically, they are $\mathbf{W}_{\mathbf{T}}^{\mathbf{ali}}$ and $\mathbf{W}_{\mathbf{S}}^{\mathbf{ali}}$ in alignment, while $\mathbf{W}_{\mathbf{T}}^{\mathbf{ver}}$ and $\mathbf{W}_{\mathbf{S}}^{\mathbf{ver}}$ in verification. As analyzed above, in the distillation of alignment and verification, teacher and student networks will be initialized by $\mathbf{W}_{\mathbf{T}}^{\mathbf{cls}}$ and $\mathbf{W}_{\mathbf{S}}^{\mathbf{cls}}$ respectively.

## 4.2 TARGET SELECTION

Based on the initialization, the second step is to select appropriate targets in the teacher network for distillation. One problem is that non-classification tasks have their own task-specific targets, but given the additional soft predictions $\mathbf{P}_{\mathbf{T}}^{\tau}$, which one should we use? To be clear, we first propose the general distillation for non-classification tasks as follows:

$$L(\mathbf{W}_{\mathbf{S}}|\mathbf{W}_{\mathbf{S}}^{\mathbf{cls}}) = \Phi\left(\mathbf{W}_{\mathbf{S}}, \mathbf{y}\right) + \alpha H(\mathbf{P}_{\mathbf{S}}^{\tau}, \mathbf{P}_{\mathbf{T}}^{\tau}) + \beta \Psi\left(\mathbf{K}_{\mathbf{S}}, \mathbf{K}_{\mathbf{T}}\right), \tag{5}$$

where $\mathbf{W}_{\mathbf{S}}$ and $\mathbf{y}$ denote the task-specific network parameter and label respectively. $\Phi\left(\mathbf{W}_{\mathbf{S}}, \mathbf{y}\right)$ is the task-specific loss function, and $\Psi\left(\mathbf{K}_{\mathbf{S}}, \mathbf{K}_{\mathbf{T}}\right)$ is the task-specific distillation term with the targets selected as $\mathbf{K}_{\mathbf{T}}$ and $\mathbf{K}_{\mathbf{S}}$ in teacher and student networks. Besides, $\alpha$ and $\beta$ are the balancing terms between classification and non-classification tasks. In Eqn.(5), the above problem has become how to set $\alpha$ and $\beta$ for a given non-classification task. In the following two parts, we will give some discussions on two tasks: face alignment and verification.

### 4.2.1 ALIGNMENT

The task of face alignment is to locate the key-point locations for each image. Without loss of generality, there is no any identity label, but only the keypoint locations for each image. Face alignment is usually considered as a regression problem(Wu et al., 2015), thus we train the teacher network with optimizing the Euclidean loss:

$$L(\mathbf{W}_{\mathbf{T}}^{\mathbf{ali}}|\mathbf{W}_{\mathbf{T}}^{\mathbf{cls}}) = \left\|\mathbf{R}_{\mathbf{T}} - \mathbf{y}^{\mathbf{ali}}\right\|^{2}, \tag{6}$$

wherein $\mathbf{R}_{\mathbf{T}}$ is the regression prediction of the teacher network and $\mathbf{y}^{\mathbf{ali}}$ is the regression label. In distillation, except for the available soft predictions $\mathbf{P}_{\mathbf{T}}^{\tau}$(classification target), another one is the task-specific target that can be the hidden layer $\mathbf{K}_{\mathbf{T}}$(Luo et al., 2016), and it satisfies $\mathbf{R}_{\mathbf{T}} = fc\left(\mathbf{K}_{\mathbf{T}}\right)$ with $fc$ being a fully-connected mapping.

In face classification, the key in distinguishing different identities is the appearance around the key-points such as shape and color, but the difference of key-point locations for different identities is small. As a result, face identity is not the main influencing factor for these locations, but it is still related as different identities may have slightly different locations. Instead, pose and viewpoint

variations have a much larger influence. Therefore, in face alignment, the hidden layer is preferred for distillation, which gives Eqn.(7) by setting $\alpha < \beta$, as shown in Fig.1(b).

$$L(\mathbf{W_S^{ali}}|\mathbf{W_S^{cls}}) = \left\|\mathbf{R_S} - \mathbf{y^{ali}}\right\|^2 + \alpha H(\mathbf{P_S^\tau}, \mathbf{P_T^\tau}) + \beta\|\mathbf{K_S} - \mathbf{K_T}\|^2. \tag{7}$$

### 4.2.2 Verification

The task of face verification is to determine if two images belong to the same identity. In verification, triplet loss(Schroff et al., 2015) is a widely used metric learning method(Schroff et al., 2015), and we take it for model distillation. Without loss of generality, we have the same identity labels as in classification, then the teacher network can be trained as

$$L(\mathbf{W_T^{ver}}|\mathbf{W_T^{cls}}) = \left[\|\mathbf{K_T^a} - \mathbf{K_T^p}\|^2 - \|\mathbf{K_T^a} - \mathbf{K_T^n}\|^2 + \lambda\right]_+, \tag{8}$$

where $\mathbf{K_T^a}$, $\mathbf{K_T^p}$ and $\mathbf{K_T^n}$ are the hidden layers for the anchor, positive and negative samples respectively, *i.e.,* $a$ and $p$ have the same identity, while $a$ and $n$ come from different identities. Besides, $\lambda$ controls the margin between positive and negative pairs.

Similar to face alignment, we consider the hidden layer $\mathbf{K_T}$ and soft prediction $\mathbf{P_T^\tau}$ as two possible targets in distillation. In fact, classification focuses on the difference of identities, *i.e.* the inter-class relation, and this relation can help a lot in telling if two image have the same identity. As a result, classification can be beneficial to boost the performance of verification. Therefore, in face verification, the soft prediction is preferred for distillation, which gives the following loss function by setting $\alpha > \beta$, as shown in Fig.1(c).

$$L(\mathbf{W_S^{ver}}|\mathbf{W_S^{cls}}) = \left[\|\mathbf{K_S^a} - \mathbf{K_S^p}\|^2 - \|\mathbf{K_S^a} - \mathbf{K_S^n}\|^2 + \lambda\right]_+ + \alpha H(\mathbf{P_S^\tau}, \mathbf{P_T^\tau}) + \beta\|\mathbf{K_S} - \mathbf{K_T}\|^2. \tag{9}$$

Particularly, some studies(Wang et al., 2017) show the benefits by using additional softmax loss in Eqn.(8). For comparison, we also add the softmax loss $H(\mathbf{P_T}, \mathbf{y^{cls}})$ and $H(\mathbf{P_S}, \mathbf{y^{cls}})$ in Eqn.(8) and Eqn.(9) respectively for further enhancement.

### 4.2.3 A Short Summary

As analyzed above, $\alpha$ and $\beta$ should be set differently in the distillation of different tasks. The key is to measure the relevance of objectives between classification and non-classification tasks. For a given task, if it is highly related to the classification task, then $\alpha > \beta$ is necessary, or $\alpha < \beta$ should be set. Though this rule cannot be theoretically guaranteed, it provides some guidelines to use knowledge distillation in more non-classification tasks.

## 5 Experimental Evaluation

In this section, we give the experimental evaluation of the proposed method. We first introduce the experimental setup in detail, and then show the results of knowledge distillation in the tasks of face classification, alignment and verification.

### 5.1 Experimental Setup

***Database***: We use three popular datasets for evaluation, including CASIA-WebFace(Yi et al., 2016), CelebA(Liu et al., 2015) and MS-Celeb-1M(Guo et al., 2016). CASIA-WebFace contains 10575 people and 494414 images, while CelebA has 10177 people with 202599 images and the label of 5 key-point locations. Compared to the previous two, MS-Celeb-1M is a large-scale dataset that contains $100K$ people with $8.4$ million images. In experiments, we use CASIA-WebFace and MS-Celeb-1M for classification, CelebA for alignment and CASIA-WebFace for verification.

***Evaluation***: In all datasets, we randomly split them into $80\%$ training and $20\%$ testing samples. In classification, we evaluate the top1 accuracy based on if the identity of the maximum prediction matches the correct identity label(Krizhevsky et al., 2012), and the results on the LFW(Learned-Miller et al., 2016) database (6000 pairs) are also reported by computing the percentage of how many pairs are correctly verified. In alignment, the Normalized Root Mean Squared Error (NRMSE) is used

to evaluate alignment(Wu et al., 2015); while in verification, we compute the Euclidean distance between each pair in testing samples, and the top1 accuracy is reported based on if a test sample and its nearest sample belong to the same identity. Particularly, LFW is not used in verification because 6000 pairs are not enough to see the difference obviously for different methods.

***Teacher and Student***: To learn the large number of identities, we use ResNet-50(He et al., 2016) as the teacher network, which is deep enough to handle our problem. For student networks, given the fact that deep student networks are better for knowledge distillation(Ba & Caruana, 2014; Urban et al., 2017; Romero et al., 2015), we remain the same depth but divide the number of convolution kernels in each layer by 2, 4 and 8, which give ResNet-50/2, ResNet-50/4 and ResNet-50/8 respectively.

***Pre-processing and Training***: Given an image, we resize it to $256 \times 256$ wherein a sub-image with $224 \times 224$ is randomly cropped and flipped. Particularly, we use no mean subtraction or image whitening, as we use batch normalization right after the input data. In training, the batchsize is set to be 256, 64 and 128 for classification, alignment and verification respectively, and the Nesterov Accelerated Gradient(NAG) is adopted for faster convergence. For the learning rate, if the network is trained from scratch, $0.1$ is used; while if the network is initialized, $0.01$ is used to continue, and 30 epochs are used in each rate. Besides, in distillation, student networks are trained with the targets of the teacher network generated online, and the temperature $\tau$ and margin $\lambda$ are set to be 3 and $0.4$ by cross-validation. Finally, the balancing terms $\alpha$ and $\beta$ have many possible combinations, and we show later how to set them by an experimental trick.

***Symbols in Experiments***: (1)$Scratch$: student networks are not initialized; (2)$Pretrain$: student networks are trained with the task-specific initialization; (3)$Distill$: student networks are initialized with $\mathbf{W_S^{ls}}$; (4)$Soft$: the soft prediction $\mathbf{P_T^\tau}$; (5)$Hidden$: the hidden layer $\mathbf{K_T}$.

## 5.2 Comparison to Previous Studies

In this part, we compare the initialization trick to previous studies in classification. Table.1 shows the comparison of different targets and initializations. It can be observed from the first table that without any initialization, soft predictions achieve the best distillation performance, *i.e.*, $61.27\%$. Based on the best target, the second table gives the results of different initializations in distillation. We see that our full initialization obtains the best accuracy of $75.06\%$, which is much higher than other methods, *i.e.*, $10\%$ and $5\%$ higher than the $Scratch$ and Fitnet(Romero et al., 2015). These results show that the full initialization of student networks can give the highest transferability in classification, and also demonstrates the effectiveness of this simple trick.

Table 1: The comparison to previous studies with different initializations and targets. Results are given on CASIA-WebFace.

| ***CASIA-WebFace*** | | ResNet-50/8 with Different Targets | | |
|---|---|---|---|---|
| | Initialization | | Targets | |
| Top1 acc(%) | Scratch | Hidden (Luo et al., 2016) | Logits (Sau & Balasubramanian, 2017) | Soft (Hinton et al., 2014) |
| | | 60.08 | 59.77 | ***61.27*** |

| ***CASIA-WebFace*** | | ResNet-50/8 with Different Initializations | | |
|---|---|---|---|---|
| | Target | | Initialization | |
| Top1 acc(%) | $\mathbf{P_T^\tau}$ | Scratch (Hinton et al., 2014) | Fitnet (Ba & Caruana, 2014) | Ours |
| | | 64.49 | 69.88 | ***75.06*** |

## 5.3 Face Classification

Base on the best initialization and target, Table.2 shows the distillation results of face classification on CASIA-WebFace and MS-Celeb-1M, and we have three main observations. Firstly, the student networks trained with full initialization can obtain large improvements over the ones trained from scratch, which further demonstrates the effectiveness of the initialization trick in large-scale cases. Secondly, some student networks can be competitive to the teacher network or even exceed the teacher one by a large margin, *e.g.* in the CASIA-WebFace database, ResNet-50/4 can be competitive to the

teacher network, while ResNet-50/2 is about $3\%$ higher than the teacher one in the top1 accuracy. Finally, in the large-scale MS-Celeb-1M, student networks cannot exceed the teacher network but only be competitive, which shows that the knowledge distillation is still challenging for a large number of identities.

Table 2: The top1 and LFW accuracy of knowledge distillation in classification. Results are obtained on CASIA-WebFace and MS-Celeb-1M.

| CASIA-WebFace | Teacher Network | Student Network | | | |
|---|---|---|---|---|---|
| | ResNet-50 | Initialization | ResNet-50/2 | ResNet-50/4 | ResNet-50/8 |
| Top1 acc(%) | 88.61 | Scratch(Hinton et al., 2014) | 82.25 | 79.36 | 66.12 |
| | | Ours | *91.01* | 87.21 | 75.06 |
| LFW acc(%) | 97.67 | Scratch(Hinton et al., 2014) | 97.27 | 96.7 | 95.12 |
| | | Ours | *98.2* | 97.57 | 96.18 |
| **MS-Celeb-1M** | Teacher Network | Student Network | | | |
| | ResNet-50 | Initialization | ResNet-50/2 | ResNet-50/4 | ResNet-50/8 |
| Top1 acc(%) | 90.53 | Scratch(Hinton et al., 2014) | 84.59 | 81.94 | 57.84 |
| | | Ours | *88.38* | 85.26 | 70.98 |
| LFW acc(%) | 99.11 | Scratch(Hinton et al., 2014) | 98.61 | 98.03 | 96.33 |
| | | Ours | *98.88* | 98.18 | 96.98 |

## 5.4 FACE ALIGNMENT

In this part, we give the evaluation of distillation in face alignment. Table.3 shows the distillation results of ResNet-50/8 with different initializations and targets on CelebA. The reason we only consider ResNet-50/8 is that face alignment is a relatively easy problem and most studies use shallow and small networks, thus a large compression rate is necessary for the deep ResNet-50. One important thing is how to set $\alpha$ and $\beta$ in Eqn.(7). As there are many possible combinations, we use a simple trick by measuring their individual influence and discard the target with the negative impact by setting $\alpha = 0$ or $\beta = 0$; while if they both have positive impacts, $\alpha > 0, \beta > 0$ should be set to keep both targets in distillation.

As shown in Table.3, when the initializations of $Pretrain$ and $Distill$ are used, $\alpha = 1, \beta = 0$(soft prediction) always decreases performance, while $\alpha = 0, \beta = 1$(hidden layer) gets consistent improvements, which implies that the hidden layer is preferred in the distillation of face alignment. It can be observed in Table.3 that $Distill$ has a lower error rate than $Pretrain$, which shows that $\mathbf{W_S^{cls}}$ has higher transferability on high-level representation than the task-specific initialization. Besides, the highest distillation performance $3.21\%$ is obtained with $Distill$ and $\alpha = 0, \beta = 1$, and it can be competitive to the one of the teacher network($3.02\%$).

Table 3: The NRMSE(%) of knowledge distillation in face alignment with different initializations and targets. Results are obtained on CelebA.

| CelebA | Teacher Network | Student Network | | | | |
|---|---|---|---|---|---|---|
| | ResNet-50 | Network | Initialization | Targets | | |
| | | | | $\alpha = 0, \beta = 0$ | $\alpha = 0, \beta = 1$ | $\alpha = 1, \beta = 0$ |
| NRMSE(%) | 3.02 | ResNet-50/8 | Pretrain | 3.36 | *3.24* | 3.60 |
| | | | Distill | 3.29 | *3.21* | 3.54 |

## 5.5 FACE VERIFICATION

In this part, we give the evaluation of distillation in face verification. Similar to alignment, we select $\alpha$ and $\beta$ in the same way. Table.4 shows the verification results of different initializations and targets on CASIA-WebFace, and the results are given by Eqn.(9). It can be observed that no matter which student network or initialization is used, $\alpha = 0, \beta = 1$(hidden layer) always decreases the baseline performance, while $\alpha = 1, \beta = 0$(soft prediction) remains almost the same. As a result, we discard the hidden layer and only use the soft prediction.

One interesting observation in Table.4 is that $\alpha = 0, \beta = 0$ always obtains the best performance, and the targets do not work at all. One possible reason is that the target in classification is not confident, *i.e.*, the top1 accuracy of ResNet-50 in classification is only $88.61\%$. To improve the classification ability, we add additional softmax loss in Eqn.(8) and Eqn.(9), and the results are shown in Table.5. We see that the accuracy of ResNet-50/2 and ResNet-50/4 has obtained remarkable improvements, which implies that the classification targets that are not confident enough cannot help the distillation. But with the additional softmax loss, the student work can adjust the learning by identity labels. As a result, $\alpha = 1, \beta = 0$ can get the best performance, which is even much higher than the teacher network, *e.g.*, $79.96\%$ of ResNet-50/2 with $Distill$ and $\alpha = 1, \beta = 0$.

Table 4: The top1 accuracy of knowledge distillation with single triplet loss in verification. Results are obtained on CASIA-WebFace.

| *CASIA-WebFace* | Teacher Network | | Student Network | | | |
|---|---|---|---|---|---|---|
| | ResNet-50 | Network | Initialization | | Targets | |
| Top1 acc(%) | | | | $\alpha = 0, \beta = 0$ | $\alpha = 0, \beta = 1$ | $\alpha = 1, \beta = 0$ |
| | 73.81 | ResNet-50/2 | Pretrain | 63.98 | 60.66 | 66.50 |
| | | | Distill | *71.29* | 68.74 | 71.23 |

| *CASIA-WebFace* | Teacher Network | | Student Network | | | |
|---|---|---|---|---|---|---|
| | ResNet-50 | Network | Initialization | | Targets | |
| Top1 acc(%) | | | | $\alpha = 0, \beta = 0$ | $\alpha = 0, \beta = 1$ | $\alpha = 1, \beta = 0$ |
| | 73.81 | ResNet-50/4 | Pretrain | 61.74 | 61.71 | 62.64 |
| | | | Distill | *68.17* | 66.74 | 68.12 |

| *CASIA-WebFace* | Teacher Network | | Student Network | | | |
|---|---|---|---|---|---|---|
| | ResNet-50 | Network | Initialization | | Targets | |
| Top1 acc(%) | | | | $\alpha = 0, \beta = 0$ | $\alpha = 0, \beta = 1$ | $\alpha = 1, \beta = 0$ |
| | 73.81 | ResNet-50/8 | Pretrain | 51.03 | 49.19 | 51.76 |
| | | | Distill | *56.69* | 53.99 | 56.52 |

Table 5: The top1 accuracy of knowledge distillation with joint triplet loss and softmax loss in verification. Results are obtained on CASIA-WebFace.

| *CASIA-WebFace* | Teacher Network | | Student Network | | | |
|---|---|---|---|---|---|---|
| | ResNet-50 | Network | Initialization | | Targets | |
| Top1 acc(%) | | | | $\alpha = 0, \beta = 0$ | $\alpha = 0, \beta = 1$ | $\alpha = 1, \beta = 0$ |
| | 74.16 | ResNet-50/2 | Pretrain | 72.38 | 70.54 | 73.62 |
| | | | Distill | 79.51 | 77.63 | *79.96* |

| *CASIA-WebFace* | Teacher Network | | Student Network | | | |
|---|---|---|---|---|---|---|
| | ResNet-50 | Network | Initialization | | Targets | |
| Top1 acc(%) | | | | $\alpha = 0, \beta = 0$ | $\alpha = 0, \beta = 1$ | $\alpha = 1, \beta = 0$ |
| | 74.16 | ResNet-50/4 | Pretrain | 66.64 | 65.08 | 68.24 |
| | | | Distill | 72.01 | 70.31 | *72.82* |

| *CASIA-WebFace* | Teacher Network | | Student Network | | | |
|---|---|---|---|---|---|---|
| | ResNet-50 | Network | Initialization | | Targets | |
| Top1 acc(%) | | | | $\alpha = 0, \beta = 0$ | $\alpha = 0, \beta = 1$ | $\alpha = 1, \beta = 0$ |
| | 74.16 | ResNet-50/8 | Pretrain | 51.86 | 51.43 | 53.45 |
| | | | Distill | 57.66 | 56.87 | *57.78* |

# 6 Conclusion

In this paper, we take face recognition as a breaking point, and propose the knowledge distillation on two non-classification tasks, including face alignment and verification. We extend the previous distillation framework by transferring the distilled knowledge from face classification to face alignment and verification. By selecting appropriate initializations and targets, the distillation on non-classification tasks can be easier. Besides, we also give some guidelines for target selection on non-classification tasks, and we hope these guidelines can be helpful for more tasks. Experiments on the datasets of CASIA-WebFace, CelebA and large-scale MS-Celeb-1M have demonstrated the effectiveness of the proposed method, which gives the student networks that can be competitive or exceed the teacher network under appropriate compression rates. In addition, we use a common initialization trick to further improve the distillation performance of classification, and this can boost the distillation on non-classification tasks. Experiments on CASIA-WebFace have demonstrated the effectiveness of this simple trick.

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
