# OpenReview forum: "Model Distillation with Knowledge Transfer from Face Classification to Alignment and Verification"
_ICLR.cc/2018/Conference — Reject_

### Official Review · AnonReviewer1 · 2017-11-26
**Limited scope and contribution**

**Rating:** 3
**Confidence:** 4

**Review:**

The paper proposes knowledge distillation on two very specific non-classification tasks. I find the scope of the paper is quite limited and the approach seems hard to generalize to other tasks. There is also very limited technical contribution. I think the paper might be a better fit in conferences on faces such as FG.

Pros:
1. The application of knowledge distillation in face alignment is interesting.

Cons:
1. The writing of the paper can be significantly improved. The technical description is unclear.
2. The method has two parameters \alpha and \beta, and Section 4.2.3. mentions the key is to measure the relevance of tasks. It seems to me defining the relevance between tasks is quite empirical and often confusing. How are they actually selected in the experiments? Sometimes alpha=0, beta=0 works the best which means the added terms are useless?
3. The paper works on a very limited scope of face alignment. How does the proposed method generalize to other tasks?

---

### Official Review · AnonReviewer3 · 2017-11-29
**This paper proposed to transfer the classifier from the model for face classification to the task of alignment and verification. The problem setting is interesting and valuable, however, the contribution is not clearly demonstrated.**

**Rating:** 5
**Confidence:** 5

**Review:**

This paper proposed to transfer the classifier from the model for face classification to the task of alignment and verification. The problem setting is interesting and valuable, however, the contribution is not clearly demonstrated.

Specifically, it proposed to utilize the teacher model from classification to other tasks, and proposed a unified objective function to model the transferability as shown in Equation (5). The two terms in (5), (7) and (9) are used to transfer the knowledge from the teacher model. It maybe possible to claim that the different terms may play different roles for different tasks. However, there should be some general guidelines for choosing these different terms for regularization, rather than just make the claim purely based on the final results. In table 4 and table 5, the results seem to be not so consistent for using the distillation loss. The author mentioned that it is due to the weak teacher model. However, the teacher model just differs in performance with around 3% in accuracy. How could we define the “good” or “bad” of a teacher model for model distillation/transfer?

Besides, it seems that the improvement comes largely from the trick of initialization as mentioned in Section 3.2. Hence, it is still not clear which parts contribute to the final performance improvements. It could be better if the authors can report the results from each of the components together.

 The authors just try the parameter (\alpha, \beta) to be (0,0), (1,0), (0,1) and (1,1). I think the range for both values could be any positive real value, and how about the performance for other sets of combinations, like (0.5, 0.5)?

---

### Official Review · AnonReviewer2 · 2017-12-01
**manuscript needs significant revisions**

**Rating:** 3
**Confidence:** 4

**Review:**

Summary:
The manuscript presents experiments on distilling knowledge from a face classification model to student models for face alignment and verification. By selecting a good initialization strategy and guidelines for selecting appropriate targets for non-classification tasks, the authors achieve improved performance, compared to networks trained from scratch or with different initialization strategies.

Review:
The paper seems to be written in a rush.
I am not sure about the degree of novelty, as pretraining with domain-related data instead of general-purpose ImageNet data has been done before, Liu et al. (2014), for example pretrain a CNN on face classification to be used for emotion recognition. Admitted, knowledge transfer from classification to regression and retrieval tasks is not very common yet, except via pretraining on ImageNet, followed by fine-tuning on the target task.
My main concern is with the presentation of the paper. It is very hard to follow! Two reasons are that it has too many grammatical mistakes and that very often a “simple trick” or a “common trick” is mentioned instead of using a descriptive name for the method used.

Here are a few points that might help improving the work:
1) Many kind of empty phrases are repeated all over the paper, e.g. the reader is teased with mention of a “simple trick” or a “common trick”. I don’t think the phrase “breaking point”, that is repeated a couple of times, is correctly used (see https://www.merriam-webster.com/dictionary/breaking%20point for a defininition).
2) Section 4.1 does not explain the initialization but just describes motivation and notation.
3) Clarity of the approach: Using the case of alignment as an example, do you first pretrain both the teacher and student on classification, then finetune the teacher on alignment before the distillation step?
4) Table 1 mentions Fitnets, but cites Ba & Caruana (2014) instead of Romero et al. (2015)
5) The “experimental trick” you mention for setting alpha and beta, seems to be just validation, comparing different settings and picking the one yielding the highest improvements. On what partition of the data are you doing this hyperparameter selection?
6) The details of the architectures are missing, e.g. exactly what changes do you make to the architecture, when you change the task from classification to alignment or verification? What exactly is the “hidden layer” in that architecture?
7) Minor: Usually there is a space before parentheses (many citations don’t have one)

In its current form, I cannot recommend the manuscript for acceptance. I get the impression that the experimental work might be of decent quality, but the manuscript fails to convey important details of the method, of the experimental setup and in the interpretation of the results. The overall quality of the write-up has to be significantly improved.

References:
Liu, Mengyi, Ruiping Wang, Shaoxin Li, Shiguang Shan, Zhiwu Huang, and Xilin Chen. "Combining multiple kernel methods on riemannian manifold for emotion recognition in the wild." In Proceedings of the 16th International Conference on Multimodal Interaction, pp. 494-501. ACM, 2014.

---

### Decision · Program_Chairs · 2018-01-29
**ICLR 2018 Conference Acceptance Decision**

**Decision:**

Reject

**Comment:**

The authors propose a distillation-based approach that is applied to transfer knowledge from a classification network to non-classification tasks (face alignment and verification). The writing is very imprecise - for instance repeatedly referring to a 'simple trick' rather than actually defining the procedure - and the method is described in very task-specific ways that make it hard to understand how or whether it would generalize to other problems.